# Magnetically Induced Carrier Distribution in a Composite Rod of Piezoelectric Semiconductors and Piezomagnetics

**DOI:** 10.3390/ma13143115

**Published:** 2020-07-13

**Authors:** Guolin Wang, Jinxi Liu, Wenjie Feng, Jiashi Yang

**Affiliations:** 1School of Civil Engineering, Shijiazhuang Tiedao University, Shijiazhuang 050043, China; wanggl@stdu.edu.cn; 2Hebei Key Laboratory of Mechanics of Intelligent Materials and Structures, Shijiazhuang Tiedao University, Shijiazhuang 050043, China; wjfeng9999@126.com; 3Department of Engineering Mechanics, Shijiazhuang Tiedao University, Shijiazhuang 050043, China; 4State Key Laboratory of Mechanical Behavior and System Safety of Traffic Engineering Structures, Shijiazhuang Tiedao University, Shijiazhuang 050043, China; 5Department of Mechanical and Materials Engineering, University of Nebraska-Lincoln, Lincoln, NE 68588, USA

**Keywords:** piezomagnetic, piezoelectric semiconductor, carrier tuning, applied magnetic field

## Abstract

In this work, we study the behavior of a composite rod consisting of a piezoelectric semiconductor layer and two piezomagnetic layers under an applied axial magnetic field. Based on the phenomenological theories of piezoelectric semiconductors and piezomagnetics, a one-dimensional model is developed from which an analytical solution is obtained. The explicit expressions of the coupled fields and the numerical results show that an axially applied magnetic field produces extensional deformation through piezomagnetic coupling, the extension then produces polarization through piezoelectric coupling, and the polarization then causes the redistribution of mobile charges. Thus, the composite rod exhibits a coupling between the applied magnetic field and carrier distribution through combined piezomagnetic and piezoelectric effects. The results have potential applications in piezotronics when magnetic fields are relevant.

## 1. Introduction

Piezoelectric materials may be dielectrics or semiconductors. In piezoelectric semiconductors, mechanical fields interact with mobile charges through the electric fields accompanying the mechanical fields produced via piezoelectric couplings. Since the 1960s, there have been efforts on developing piezoelectric semiconductor acoustoelectric wave devices based on these couplings [1]. Relatively recently, various piezoelectric semiconductor materials and structures have been synthesized, such as fibers, tubes, belts, spirals, and films using the so-called third-generation semiconductors, such as ZnO and MoS_2_, which are piezoelectric [2]. These materials have great potentials for broad applications in electronics and phototronics in the form of single structures or arrays [3,4], sensors [5], electro- and photochemical applications [6], optoelectronics [7], and nanogenerators [8,9]. These relatively recent developments have formed new research areas called piezotronics and piezo-phototronics.

If a piezomagnetic material is attached to a piezoelectric semiconductor, the resulting composite structure deforms in a magnetic field due to piezomagnetic coupling. The deformation then produces electric polarization and motion or redistribution of mobile charges in the piezoelectric semiconductor [10]. This effect has been explored for applications in nanogenerators [11,12], optical devices [13,14], transistors [15], magnetic recording devices [16], and sensors [11]. Because of these applications, there is an emerging and growing need to study the coupling behavior of the composite structures of piezoelectric semiconductors and piezomagnetics.

In this paper, we study multi-field interactions in piezomagnetic–piezoelectric semiconductor composite structures through a theoretical analysis of the basic problem of a rod in extensional deformation under an axial magnetic field. The analysis is simple, which allows us to show the physics involved and the roles of various physical and geometric parameters explicitly. The macroscopic theories for piezoelectric semiconductors and piezomagnetics are summarized in Section 2. A one-dimensional model for the extensional deformation of a composite rod is developed in Section 3, along with an analytical solution in Section 4. Numerical results and discussions are presented in Section 5, with a few conclusions in Section 6.

## 2. Governing Equations

We consider the structure of the composite rod shown in Figure 1. It consists of a piezoelectric semiconductor layer “(1)” and two identical piezomagnetic layers “(2).” It is under an axial magnetic field *H*_3_, which causes axial extension of the rod through the piezomagnetic constant *h*_33_. If the direction of M is in the *x*_2_ direction perpendicular to the piezomagnetic layers, the extension can be produced in the rod under a transverse magnetic field in a similar way through the piezomagnetic constant *h*_31_. The case of piezoelectric/piezomagnetic dielectric composites have been well studied, e.g., [17,18,19,20,21]. More references can be found in a review [22]. Our composite rod differs from the literature in that the piezoelectric layer in Figure 1 is a semiconductor.

The basic behaviors of the materials of the structure in Figure 1 can be described by the following equations of piezomagnetic and piezoelectric semiconductors [23,24] in a Cartesian coordinate system *x_j_* (*j* = 1, 2, 3):(1)$$\frac{\partial T_{ji}}{\partial x_{j}}=\rho \frac{\partial^{2}u_{i}}{\partial t^{2}},$$
(2)$$\frac{\partial D_{i}}{\partial x_{i}}=q(p-n+N_{D}^{+}-N_{A}^{-}),$$
(3)$$\frac{\partial B_{i}}{\partial x_{i}}=0,$$
(4)$$\frac{\partial J_{i}^{p}}{\partial x_{i}}=-q\frac{\partial p}{\partial t},$$
(5)$$\frac{\partial J_{i}^{n}}{\partial x_{i}}=q\frac{\partial n}{\partial t},$$
where $T_{ij}^{}$ is the stress tensor, ρ is the mass density, $u_{i}$ is the mechanical displacement vector, $D_{i}$ is the electric displacement vector, $q=1.6\times 10^{-19}$C is the elementary charge, *p* and *n* are the concentrations of holes and electrons, $N_{D}^{+}$ and $N_{A}^{-}$ are the concentrations of ionized donors and accepters, which are assumed to be uniform in this paper, $J_{i}^{p}$ and $J_{i}^{n}$ are the hole and electron current densities, and $B_{i}$ is the magnetic flux or induction vector. In the above equations, repeated subscripts are summed from 1 to 3. Equation (1) is the stress equation of motion (Newton’s law). Equation (2) is the charge equation of electrostatics. Equation (3) is the Gauss equation for the magnetic induction. Equations (4) and (5) are the conservation of charge for holes and electrons (continuity equations). The related constitutive relations describing material behaviors are(6)$$\begin{array}{l} T_{ij}=c_{ijkl}S_{kl}-e_{kij}E_{k}-h_{kij}H_{k},\\ D_{i}=e_{ikj}S_{kl}+\varepsilon_{ik}E_{k}+\alpha_{ik}H_{k},\\ B_{i}=h_{ikl}S_{kl}+\alpha_{ik}E_{k}+\mu_{ik}H_{k}, \end{array}$$
(7)$$J_{i}^{p}=qp\mu_{ij}^{p}E_{j}-qD_{ij}^{p}\frac{\partial p}{\partial x_{j}},$$
(8)$$J_{i}^{n}=qn\mu_{ij}^{n}E_{j}+qD_{ij}^{n}\frac{\partial n}{\partial x_{j}},$$
where $S_{ij}$ is the strain tensor, $E_{i}$ is the electric field vector, and $H_{i}$ is the magnetic field vector. Equation (6) are the constitutive relations for piezoelectrics and piezomagnetics. Equations (7) and (8) are the constitutive relations for the current densities, including both the drift and diffusion currents. $c_{ijkl}$ is the elastic stiffness. $e_{ijk}$ is the piezoelectric constant that describes the coupling between mechanical and electric fields. $h_{ijk}$ is the piezomagnetic constant that describes the coupling between mechanical and magnetic fields. $\varepsilon_{ij}$ is the dielectric constant. $\alpha_{ij}$ is the magnetoelectric constant. $\mu_{ij}$ is the magnetic permeability. $\mu_{ij}^{p}$ and $\mu_{ij}^{n}$ are the carrier mobilities. $D_{ij}^{p}$ and $D_{ij}^{n}$ are the carrier diffusion constants. The strain-displacement and field-potential relations are(9)$$S_{ij}=\frac{1}{2}\left(\frac{\partial u_{j}}{\partial x_{i}}+\frac{\partial u_{i}}{\partial x_{j}}\right),$$
(10)$$E_{i}=-\frac{\partial \varphi }{\partial x_{i}},$$
(11)$$H_{i}=-\frac{\partial \psi }{\partial x_{i}}$$
where *φ* is the electric potential and *ψ* is the magnetic potential. For the purpose of this paper, the following linearized version is sufficient. Let
(12)$$p=p_{0}+\Delta p,\;n=n_{0}+\Delta n,$$
where
(13)$$p_{0}=N_{A}^{-},\;n_{0}=N_{D}^{+}.$$
Then Equations (2), (4) and (5) become
(14)$$\frac{\partial D_{i}}{\partial x_{i}}=q(\Delta p-\Delta n),$$
(15)$$\begin{array}{l} q\frac{\partial }{\partial t}(\Delta p)=-\frac{\partial J_{i}^{p}}{\partial x_{i}},\\ q\frac{\partial }{\partial t}(\Delta n)=\frac{\partial J_{i}^{n}}{\partial x_{i}}. \end{array}$$
For small Δp and Δn, we linearize Equations (7) and (8) as
(16)$$\begin{array}{l} J_{i}^{p}=qp_{0}\mu_{ij}^{p}E_{j}-qD_{ij}^{p}\frac{\partial (\Delta p)}{\partial x_{j}},\\ J_{i}^{n}=qn_{0}\mu_{ij}^{n}E_{j}+qD_{ij}^{n}\frac{\partial (\Delta n)}{\partial x_{j}}. \end{array}$$
In the reference state, *p* = *p*_0_, *n* = *n*_0_, and all other fields vanish. The above equations are applicable to each component phase of the composite structure in Figure 1 as special cases. They have been used to study thickness vibration of plates [25,26], wave propagation [27,28,29,30,31,32], fields near cracks [33,34,35], extension of rods [36,37,38,39], bending of beams [40,41,42,43,44,45], and fields near PN junctions [46,47,48,49] in piezoelectric semiconductors.

## 3. One-Dimensional Model for Extension

The equations in the previous section present considerable mathematical challenges. We consider thin rods and make a few approximations to simplify the problem. We assume that the following is approximately true throughout the composite rod during extension:(17)$$u_{3}=u≅u(x_{3},t),\;\varphi ≅\varphi (x_{3},t),\;\psi ≅\psi (x_{3},t),$$
which are understood to be averages of the corresponding three-dimensional fields over the cross-section of the rod. Then
(18)$$S_{33}=\frac{\partial u}{\partial x_{3}},\;E_{3}=-\frac{\partial \varphi }{\partial x_{3}},\;H_{3}=-\frac{\partial \psi }{\partial x_{3}}.$$
Consider the piezoelectric semiconductor layer first. We perform the stress relaxation for thin rods (*T*_11_ = *T*_22_ = 0) using the following relevant constitutive relations from Equation (6):(19)$$\begin{array}{l} T_{11}=c_{11}^{\left(1\right)}S_{11}+c_{12}^{\left(1\right)}S_{22}+c_{13}^{\left(1\right)}S_{33}-e_{31}^{\left(1\right)}E_{3}=0,\\ T_{22}=c_{12}^{\left(1\right)}S_{11}+c_{11}^{\left(1\right)}S_{22}+c_{13}^{\left(1\right)}S_{33}-e_{31}^{\left(1\right)}E_{3}=0, \end{array}$$
(20)$$\begin{array}{l} T_{33}=c_{13}^{\left(1\right)}S_{11}+c_{13}^{\left(1\right)}S_{22}+c_{33}^{\left(1\right)}S_{33}-e_{33}^{\left(3\right)}E_{3},\\ D_{3}=e_{31}^{\left(1\right)}(S_{11}+S_{22})+e_{33}^{\left(1\right)}S_{33}+\varepsilon_{33}^{\left(1\right)}E_{3}. \end{array}$$
Solving Equation (19) for expressions of *S*_11_ and *S*_22_, and then substituting them into Equation (20), one obtains the following constitutive relations for the extension of the piezoelectric semiconductor layer:(21)$$\begin{array}{l} T=c^{\left(1\right)}S-e^{\left(1\right)}E,\\ D=e^{\left(1\right)}S+\varepsilon^{\left(1\right)}E, \end{array}$$
where the relevant axial fields and the one-dimensional effective material constants are denoted by
(22)$$\begin{array}{l} S=S_{33},\;T=T_{33},\;\\ E=E_{3},\;D=D_{3}, \end{array}$$
(23)$$\begin{array}{l} c^{\left(1\right)}=c_{33}^{\left(1\right)}-\frac{2{\left(c_{13}^{\left(1\right)}\right)}^{2}}{c_{11}^{\left(1\right)}+c_{12}^{\left(1\right)}},\\ e^{\left(1\right)}=e_{33}^{\left(1\right)}-\frac{2c_{13}^{\left(1\right)}e_{31}^{\left(1\right)}}{c_{11}^{\left(1\right)}+c_{12}^{\left(1\right)}},\\ \varepsilon^{\left(1\right)}=\varepsilon_{33}^{\left(1\right)}+\frac{2{\left(e_{31}^{\left(1\right)}\right)}^{2}}{c_{11}^{\left(1\right)}+c_{12}^{\left(1\right)}}. \end{array}$$
We also denote the axial magnetic fields and the relevant magnetic material constant by
(24)$$B=B_{3},\;H=H_{3},\;\mu^{\left(1\right)}=\mu_{33}^{\left(1\right)}.$$
Then,
(25)$$B=\mu^{\left(1\right)}H.$$
In the thin piezoelectric semiconductor layer, we also make the following approximations:(26)$$\Delta p≅\Delta p(x_{3},t),\;\Delta n≅\Delta n(x_{3},t).$$
With Equations (16) and (26), the constitutive relations for the axial current densities in the piezoelectric semiconductor layer can be simplified as
(27)$$\begin{array}{l} J^{p}≅qp_{0}\mu^{p}E-qD^{p}\frac{\partial (\Delta p)}{\partial x_{3}},\\ J^{n}≅qn_{0}\mu^{n}E+qD^{n}\frac{\partial (\Delta n)}{\partial x_{3}}, \end{array}$$
where
(28)$$\begin{array}{l} \mu^{p}=\mu_{33}^{p},\;D^{p}=D_{33}^{p},\\ \mu^{n}=\mu_{33}^{n},\;D^{n}=D_{33}^{n}. \end{array}$$
Similarly, for the piezomagnetic layers, after the lateral stress relaxation, we have
(29)$$\begin{array}{l} T=c^{\left(2\right)}S-h^{\left(2\right)}H,\\ D=\varepsilon^{\left(2\right)}E,\\ B=h^{\left(2\right)}S+\mu^{\left(2\right)}H, \end{array}$$
where
(30)$$\begin{array}{l} c_{}^{\left(2\right)}=c_{33}^{\left(2\right)}-\frac{2{\left(c_{13}^{\left(2\right)}\right)}^{2}}{c_{11}^{\left(2\right)}+c_{12}^{\left(2\right)}},\;h^{\left(2\right)}=h_{33}^{\left(2\right)}-\frac{2c_{13}^{\left(2\right)}h_{31}^{\left(2\right)}}{c_{11}^{\left(2\right)}+c_{12}^{\left(2\right)}},\\ \varepsilon^{\left(2\right)}=\varepsilon_{33}^{\left(2\right)},\;\mu^{\left(2\right)}=\mu_{33}^{\left(2\right)}+\frac{2{\left(h_{31}^{\left(2\right)}\right)}^{2}}{c_{11}^{\left(2\right)}+c_{12}^{\left(2\right)}}. \end{array}$$
For the composite rod, the total axial force is calculated from the integration of *T*_3_ over the entire cross-section of the composite rod, which in this case takes the following form:(31)$$\begin{array}{l} \hat{T}=TA^{\left(1\right)}+TA^{\left(2\right)}\\ \;=\left(c^{\left(1\right)}S-e^{\left(1\right)}E\right)A^{\left(1\right)}+\left(c^{\left(2\right)}S-h^{\left(2\right)}H\right)A^{\left(2\right)}\\ \;=\left(c^{\left(1\right)}A^{\left(1\right)}+c^{\left(2\right)}A^{\left(2\right)}\right)S-e^{\left(1\right)}A^{\left(1\right)}E-h^{\left(2\right)}A^{\left(2\right)}H\\ \;=\hat{c}S-\hat{e}E-\hat{h}H, \end{array}$$
where
(32)$$\begin{array}{l} \hat{c}=c^{\left(1\right)}A^{\left(1\right)}+c^{\left(2\right)}A^{\left(2\right)},\\ \hat{e}=e^{\left(1\right)}A^{\left(1\right)},\;\hat{h}=h^{\left(2\right)}A^{\left(2\right)},\\ A^{\left(1\right)}=2bc,\;A^{\left(2\right)}=2bh. \end{array}$$
$A^{\left(1\right)}$ and $A^{\left(2\right)}$ are the cross-sectional areas of the piezoelectric semiconductor and piezomagnetic layers, respectively. Similarly, the total axial electric displacement and total axial magnetic induction over the cross-section of the composite rod are
(33)$$\begin{array}{l} \hat{D}=DA^{\left(1\right)}+DA^{\left(2\right)}\\ \;=\left(e^{\left(1\right)}S+\varepsilon^{\left(1\right)}E\right)A^{\left(1\right)}+\left(\varepsilon^{\left(2\right)}E\right)A^{\left(2\right)}\\ \;=e^{\left(1\right)}A^{\left(1\right)}S+\left(\varepsilon_{}^{\left(1\right)}A^{\left(1\right)}+\varepsilon_{}^{\left(2\right)}A^{\left(2\right)}\right)E\\ \;=\hat{e}S+\hat{\varepsilon }E, \end{array}$$
(34)$$\begin{array}{l} \hat{B}=BA^{\left(1\right)}+BA^{\left(2\right)}\\ \;=\left(\mu^{\left(1\right)}H\right)A^{\left(1\right)}+\left(h^{\left(2\right)}S+\mu^{\left(2\right)}H\right)A^{\left(2\right)}\\ \;=h^{\left(2\right)}A^{\left(2\right)}S+\left(\mu_{}^{\left(1\right)}A^{\left(1\right)}+\mu_{}^{\left(2\right)}A^{\left(2\right)}\right)H\\ \;=\hat{h}S+\hat{\mu }H, \end{array}$$
where
(35)$$\begin{array}{l} \hat{\varepsilon }=\varepsilon^{\left(1\right)}A^{\left(1\right)}+\varepsilon^{\left(1\right)}A^{\left(2\right)},\\ \hat{\mu }=\mu^{\left(1\right)}A^{\left(1\right)}+\mu^{\left(1\right)}A^{\left(2\right)}. \end{array}$$
For extension, the equation of motion of the rod in the axial direction can be obtained by considering a differential element of the rod with length *dx*_3_ as shown in Figure 2, which leads to [39]
(36)$$\frac{\partial \hat{T}}{\partial x_{3}}+f(x_{3},t)=2b\left(\rho^{\left(1\right)}c+\rho^{\left(2\right)}h\right)\frac{\partial^{2}u}{\partial t^{2}}.$$
where $f(x_{3},t)$ is the axial mechanical load per unit length of the rod. Similarly [39], the one-dimensional charge equation of electrostatics, the one-dimensional Gauss equation of the magnetic induction, and the one-dimensional conservation of holes and electrons of the composite rod are
(37)$$\frac{\partial \hat{D}}{\partial x_{3}}=q(\Delta p-\Delta n)A^{\left(1\right)},$$
(38)$$\frac{\partial \hat{B}}{\partial x_{3}}=0,$$
(39)$$\begin{array}{l} q\frac{\partial }{\partial t}(\Delta p)=-\frac{\partial J^{p}}{\partial x_{3}},\\ q\frac{\partial }{\partial t}(\Delta n)=\frac{\partial J^{n}}{\partial x_{3}}. \end{array}$$
Substituting Equations (27), (31), (33), and (34) into Equations (36)–(39), along with the use of Equation (18), one obtains
(40)$$\begin{array}{l} \hat{c}\frac{\partial^{2}u}{\partial x_{3}^{2}}+\hat{e}\frac{\partial^{2}\varphi }{\partial x_{3}^{2}}+\hat{h}\frac{\partial^{2}\psi }{\partial x_{3}^{2}}+f(x_{3},t)=2b\left(\rho^{\left(1\right)}c+\rho^{\left(2\right)}h\right)\frac{\partial^{2}u}{\partial t^{2}},\\ \hat{e}\frac{\partial^{2}u}{\partial x_{3}^{2}}-\hat{\varepsilon }\frac{\partial^{2}\varphi }{\partial x_{3}^{2}}=q\left(\Delta p-\Delta n\right)A^{\left(1\right)},\\ \hat{h}\frac{\partial^{2}u}{\partial x_{3}^{2}}-\hat{\mu }\frac{\partial^{2}\varphi }{\partial x_{3}^{2}}=0,\\ p_{0}\mu^{p}\frac{\partial^{2}\varphi }{\partial x_{3}^{2}}+D^{p}\frac{\partial^{2}\Delta p}{\partial x_{3}^{2}}=\frac{\partial }{\partial t}(\Delta p),\\ D^{n}\frac{\partial^{2}\Delta n}{\partial x_{3}^{2}}-n_{0}\mu^{n}\frac{\partial^{2}\varphi }{\partial x_{3}^{2}}=\frac{\partial }{\partial t}(\Delta n). \end{array}$$
This is a system of coupled linear partial differential equations for *u*, *φ*, *ψ*, Δp, and Δn.

## 4. Analytical Solution

Specifically, we investigate the static extension of a mechanically free (*f* = 0) and electrically isolated rod under a static axial magnetic field produced by a magnetic potential difference at the two ends of the rod. The rod is within |*x*_3_| < *L*. The boundary conditions are
(41)$$\begin{array}{l} \hat{T}(\pm L)=0,\;\hat{D}(\pm L)=0,\;\psi (\pm L)=\pm \psi_{0},\\ J^{n}(\pm L)=0,\;J^{p}(\pm L)=0. \end{array}$$
We are not considering carrier recombination and generation. Therefore, Δp and Δn must satisfy the following global charge conservation conditions:(42)$$\int_{-L}^{L}\Delta pdx_{3}=0,\;\int_{-L}^{L}\Delta ndx_{3}=0.$$
Only one of Equation (42) is independent. The other is implied by integrating Equation (37) between −*L* and *L* and using the boundary conditions on $\hat{D}$ in Equation (41), which implies that
(43)$$\int_{-L}^{L}q(\Delta p-\Delta n)dx_{3}=0.$$
Since there are no boundary conditions prescribed directly on the mechanical displacement and electric potential, the mechanical displacement may have an arbitrary constant representing a rigid-body translation of the rod along *x*_3_. At the same time, the electric potential may have an arbitrary constant that does not make any difference in the electric field it produces. To determine the mechanical displacement and electric potential uniquely, we set
(44)$$u_{3}(0)=0,\;\varphi (0)=0.$$
The relevant component of the polarization vector and distributed effective polarization charge can be calculated from
(45)$$\begin{array}{l} P=P_{3}=D_{3}-\varepsilon_{0}E_{3},\\ \rho^{P}=-P_{i,i}=-P_{3,3},\\ D_{3}=\hat{D}/A,\;A=A^{\left(1\right)}+A^{\left(2\right)}. \end{array}$$
The problem is time-independent. Thus, the terms on the right side of Equation (40) vanishes. Equation (40) reduces to a system of linear ordinary differential equations with constant coefficients. The solution can be obtained in a straightforward manner. The results are
(46)$$\psi =\frac{{\hat{e}}^{2}{\hat{h}}^{2}}{\tilde{c}\hat{c}\tilde{\varepsilon }\hat{\mu }}\frac{\psi_{0}}{\Delta }\sinh\left(kx_{3}\right)+k\cosh\left(kL\right)\frac{\psi_{0}}{\Delta }x_{3},$$
(47)$$u=\frac{{\hat{e}}^{2}\hat{h}}{\tilde{c}\hat{c}\tilde{\varepsilon }}\frac{\psi_{0}}{\Delta }\sinh\left(kx_{3}\right)-\frac{\hat{h}k\cosh\left(kL\right)}{\hat{c}}\frac{\psi_{0}}{\Delta }x_{3},$$
(48)$$S=\frac{{\hat{e}}^{2}\hat{h}k}{\tilde{c}\hat{c}\tilde{\varepsilon }}\frac{\psi_{0}}{\Delta }\cosh\left(kx_{3}\right)-\frac{\hat{h}k\cosh\left(kL\right)}{\hat{c}}\frac{\psi_{0}}{\Delta },$$
(49)$$\varphi =-\frac{\hat{e}\hat{h}}{\hat{c}\tilde{\varepsilon }}\frac{\psi_{0}}{\Delta }\sinh\left(kx_{3}\right),$$
(50)$$E=\frac{\hat{e}\hat{h}k}{\hat{c}\tilde{\varepsilon }}\frac{\psi_{0}}{\Delta }\cosh\left(kx_{3}\right),$$
(51)$$D=\frac{\hat{e}\hat{h}k}{\hat{c}A}\frac{\psi_{0}}{\Delta }\cosh\left(kx_{3}\right)-\frac{\hat{e}\hat{h}k\cosh\left(kL\right)}{\hat{c}A}\frac{\psi_{0}}{\Delta },$$
(52)$$P=\frac{\left(\tilde{\varepsilon }-\varepsilon_{0}A\right)\hat{e}\hat{h}k}{\hat{c}\tilde{\varepsilon }A}\frac{\psi_{0}}{\Delta }\cosh\left(kx_{3}\right)-\frac{\hat{e}\hat{h}k\cosh\left(kL\right)}{\hat{c}A}\frac{\psi_{0}}{\Delta },$$
(53)$$\rho^{P}=\frac{\left(\varepsilon_{0}A-\tilde{\varepsilon }\right)\hat{e}\hat{h}k^{2}}{\hat{c}\tilde{\varepsilon }A}\frac{\psi_{0}}{\Delta }\sinh\left(kx_{3}\right),$$
(54)$$\Delta n=-\frac{\mu^{n}n_{0}}{D^{n}}\frac{\hat{e}\hat{h}}{\hat{c}\tilde{\varepsilon }}\frac{\psi_{0}}{\Delta }\sinh\left(kx_{3}\right),$$
(55)$$\Delta p=\frac{\mu^{p}p_{0}}{D^{p}}\frac{\hat{e}\hat{h}}{\hat{c}\tilde{\varepsilon }}\frac{\psi_{0}}{\Delta }\sinh\left(kx_{3}\right),$$
where
(56)$$\begin{array}{l} k^{2}=\frac{qA^{\left(1\right)}}{\tilde{\varepsilon }}\left(\frac{\mu^{p}}{D^{p}}p_{0}+\frac{\mu^{n}}{D^{n}}n_{0}\right),\\ \tilde{\varepsilon }=\hat{\varepsilon }+\frac{{\hat{e}}^{2}}{\tilde{c}},\;\tilde{c}=\hat{c}+\frac{{\hat{h}}^{2}}{\hat{\mu }},\\ \Delta =kL\cosh\left(kL\right)+\frac{{\hat{e}}^{2}{\hat{h}}^{2}}{\tilde{c}\hat{c}\tilde{\varepsilon }\hat{\mu }}\sinh\left(kL\right). \end{array}$$

## 5. Numerical Results and Discussion

Based on the analytical solution in the previous section, the coupled fields are calculated and examined below. *n*-type ZnO is chosen as the piezoelectric semiconductor layer, while the two identical piezomagnetic layers are either CoFe_2_O_4_ or Terfenol-D. The relevant material properties are listed in Table 1.

We first examine the influence of the applied *ψ*_0_ and the initial carrier density on various fields. For the CoFe_2_O_4_/ZnO/CoFe_2_O_4_ composite rod with *L* = 0.6 µm, *h* = *c* =0.05 µm, and *b* = 0.2 µm, Figure 3 shows the axial distributions of the magnetic potential and piezomagnetically induced mechanical fields along the rod for different values of *ψ*_0_ when $n_{0}=1\times 10^{21}/m^{3}$. *ψ* is dominated by the applied *ψ*_0_ and is almost linear. To show the effect of couplings between *ψ* and other fields more clearly, in Figure 3a we plot $\Delta \psi =\psi -\psi_{0}x_{3}/L$ instead of *ψ* itself. As *ψ*_0_ increases, all fields become stronger as expected. It is well known that in the special case when the piezoelectric layer in the middle is a dielectric without semiconduction, the magnetic potential and mechanical displacement in Figure 3b are both linear functions of *x*_3_ and, at the same time, the strain in Figure 3c is a constant. Because of semiconduction, all of these fields have hyperbolic behaviors as indicated by their expressions in Equations (46)–(48), especially near the ends of the rod where *kL* is relatively large.

Figure 4 shows the electric potential, electric field, and electric displacement produced by the extensional deformation through piezoelectric coupling. It can be seen from Figure 4 that the applied *ψ*_0_ has obvious influences on these electric variables. Again, they differ from the linear fields or constants in composite structures of piezoelectric and piezomagnetic dielectrics.

Our main interest is the development of the distributions of mobile charges in Figure 5b, which shows that the applied magnetic field causes redistribution of charge carriers through combined piezomagnetic/piezoelectric couplings and semiconduction. Figure 5a,b shows that the electrons redistribute themselves in such a way that they tend to screen the effective polarization charges. The applied *ψ*_0_ used is relatively small to insure that Δ*n* is much smaller than *n*_0_, so that the assumption leading to the linearization in Equation (16) is not violated.

For the same composite rod, Figure 6, Figure 7 and Figure 8 show the effect of *n*_0_ on various fields when $\psi_{0}=10^{-4}\mathrm{AT}$. Specifically, Figure 6 shows the magnetic potential and piezomagnetically induced mechanical fields. Figure 6b indicates that *n*_0_ has almost no influence on the mechanical displacement. From Figure 6a,c as well as Figure 7, it can be seen that the absolute values of Δ*ψ*, strain, and electric potential decrease monotonically with the increase of *n*_0_, but the electric filed and electric displacement increase monotonically. In addition, the effect of *n*_0_ on these fields is relatively small near the two ends and the middle of the rod. Figure 8 shows the variations of the effective polarization charge and electron concentration perturbation. They assume maximal values at the ends of the rod.

In order to reveal the dependence of the electron concentration perturbation on the material combinations and the thickness ratio *h/c* between the piezomagnetic layers and piezoelectric semiconductor layer, we rewrite Equation (54) as
(57)$$\frac{\Delta n}{n_{0}}=-\frac{q}{k_{B}T}\gamma \psi_{0},$$
where
(58)$$\gamma =\frac{\hat{e}\hat{h}}{\hat{c}\tilde{\varepsilon }\Delta }\sinh\left(kx_{3}\right).$$
In Equation (57), the following Einstein relation has been used:(59)$$\frac{\mu^{n}}{D^{n}}=\frac{\mu^{p}}{D^{p}}=\frac{q}{k_{B}T},$$
where *T* is the absolute temperature and *k_B_* is the Boltzmann constant. *γ* describes the strength of the coupling effect of interest, i.e., the development of carrier redistribution under a magnetic field. For a given cross-section location, *γ* depends on the relevant material constants and the thickness ratio *h**/c.* It also varies with *x*_3_. Figure 9 shows the variation of *γ* with *h/c* for two material combinations, i.e., CoFe_2_O_4_/ZnO/CoFe_2_O_4_ and Terfenol-D/ZnO/Terfenol-D while *h + c* is held constant. It can be seen that for the CoFe_2_O_4_/ZnO/CoFe_2_O_4_ rod, *γ* is always less than that for Terfenol-D/ZnO/Terfenol-D rod. At the ends of the rod, *γ* has a maximum for a certain value of *h/c*. This is as expected because either the piezoelectric semiconductor layer or piezomagnetic layers cannot be too thin. Otherwise, there will be insufficient mobile charges or insufficient piezomagnetically induced deformation. Both the exact value of *h/c* for the maximal *γ* and the value of the maximal *γ* are sensitive to the component materials. Compared with Terfenol-D, CoFe_2_O_4_ has a larger *h*_33_, which increases *γ* according to Equation (58), but CoFe_2_O_4_ has a much larger *c*_33_, which lowers *γ*. The net result of these two competing effects is that the Terfenol-D/ZnO/Terfenol-D rod has a significantly larger *γ* than the CoFe_2_O_4_/ZnO/CoFe_2_O_4_ rod.

## 6. Conclusions

We have shown theoretically that in a properly constructed composite rod of piezoelectric semiconductors and piezomagnetics, an applied axial magnetic field produces a series of fields, including extensional deformation through the piezomagnetic coupling, polarization through the piezoelectric coupling, and redistribution of mobile charges because of semiconduction. The rod may be potentially used as a magnetic field sensor or magnetic field-to-current transducer. The material combination and thickness ratio between the piezomagnetic layer and piezoelectric semiconductor layer has strong influences on the strength of the coupling between the applied magnetic field and carrier redistribution. For a given material combination, there exists an optimal thickness ratio at which the coupling is the strongest. Hence, the redistribution or motion of mobile charges in the composite rod can be modulated by the applied magnetic field with proper design of the structure through materials and geometry.

## Figures and Tables

**Figure 1 materials-13-03115-f001:**
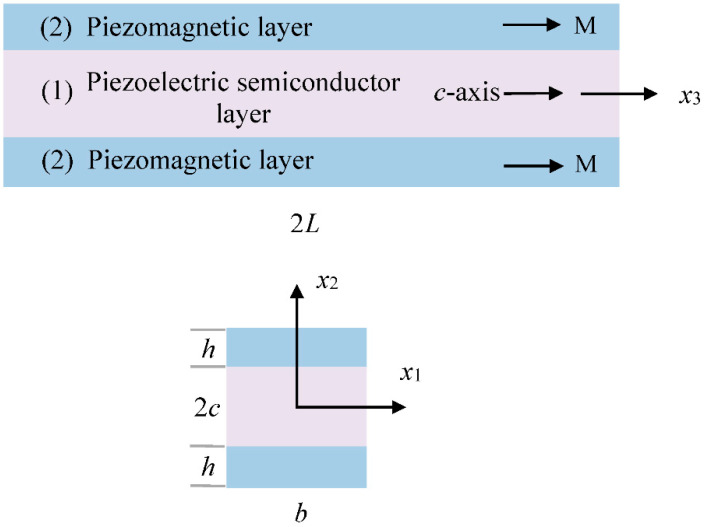
A composite rod of a piezoelectric semiconductor and piezomagnetics.

**Figure 2 materials-13-03115-f002:**
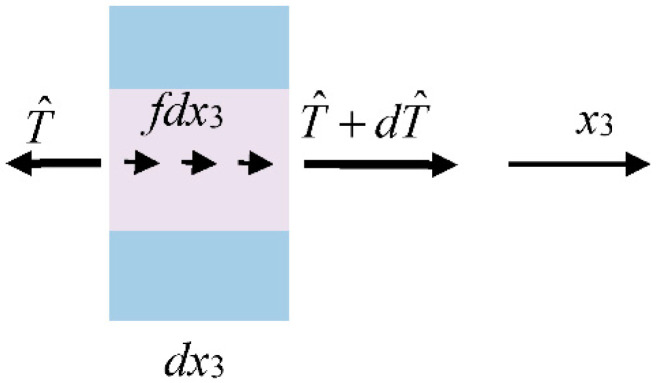
A differential element of the composite rod under mechanical loads.

**Figure 3 materials-13-03115-f003:**
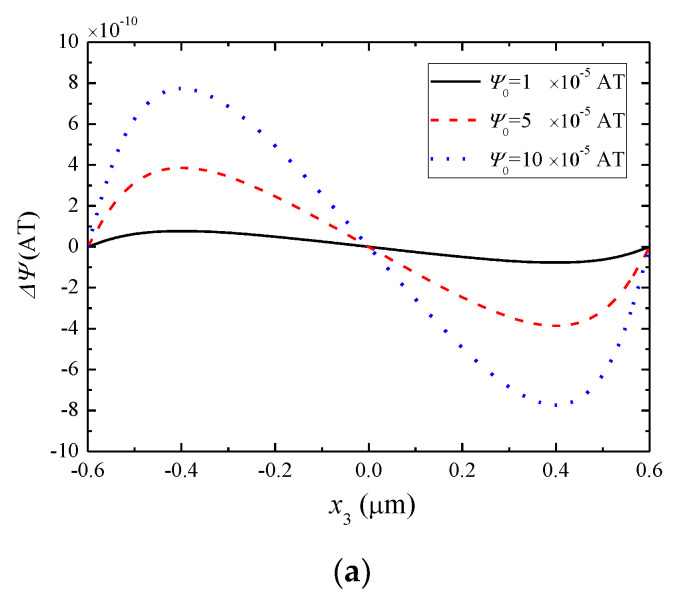
Magnetic potential and piezomagnetically induced mechanical fields under different $\psi_{0}$ when $n_{0}=1\times 10^{21}/m^{3}$. (**a**) $\Delta \psi =\psi -\psi_{0}x_{3}/L$, (**b**) mechanical displacement, (**c**) strain.

**Figure 4 materials-13-03115-f004:**
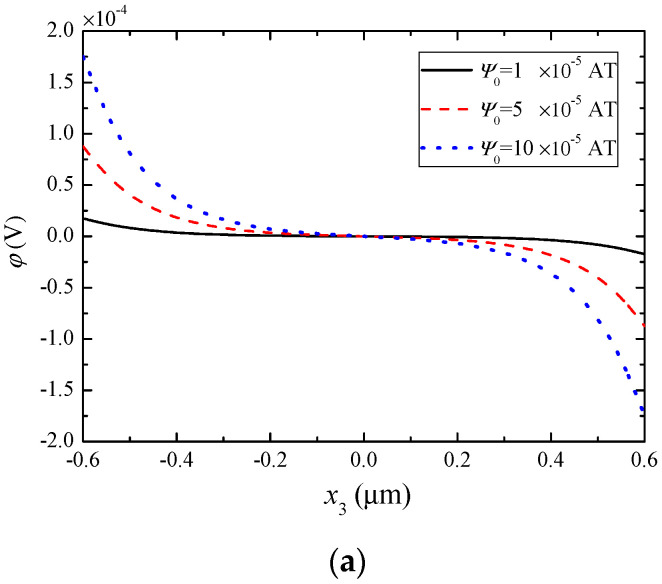
Piezoelectrically induced electric fields under different $\psi_{0}$ when $n_{0}=1\times 10^{21}/m^{3}$. (**a**) Electric potential, (**b**) electric field, (**c**) electric displacement.

**Figure 5 materials-13-03115-f005:**
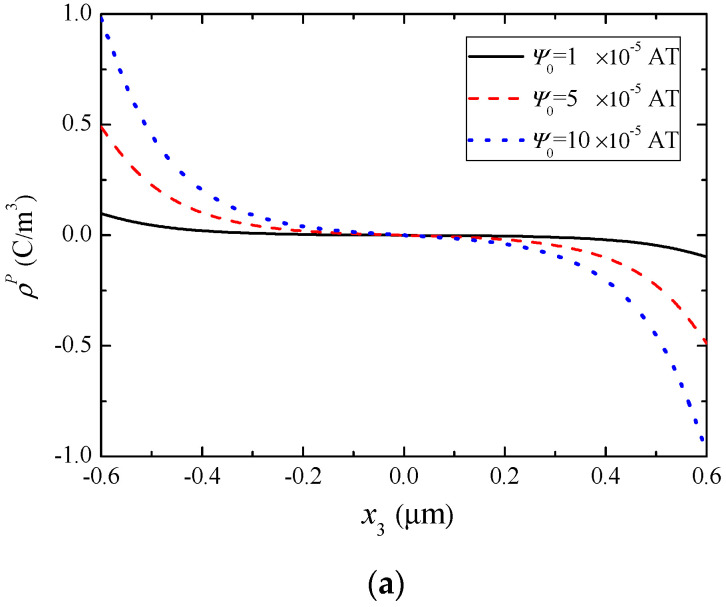
Polarization-induced charge distributions under different $\psi_{0}$ when $n_{0}=1\times 10^{21}/m^{3}$. (**a**) Effective polarization charge and (**b**) electron concentration perturbation.

**Figure 6 materials-13-03115-f006:**
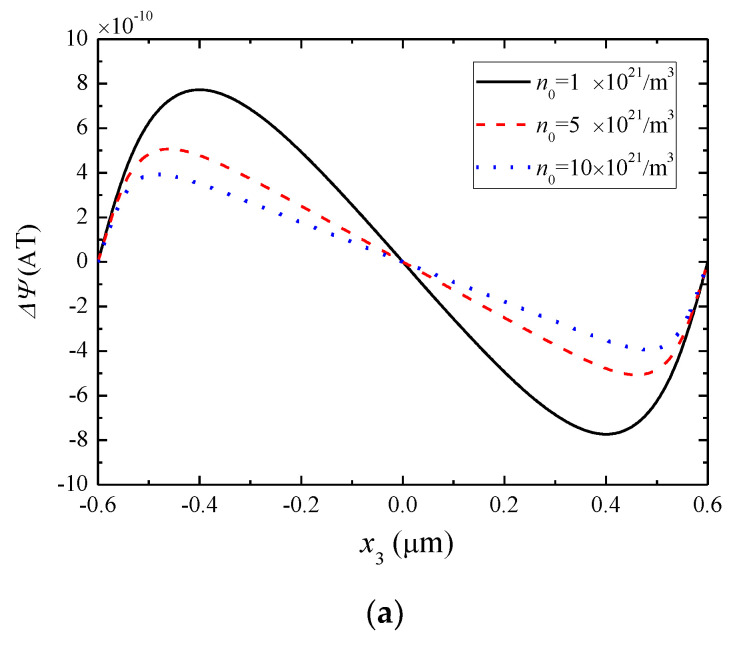
Magnetic potential and piezomagnetically induced mechanical fields for different *n*_0_ when $\psi_{0}=10^{-4}\mathrm{AT}$. (**a**) $\Delta \psi =\psi -\psi_{0}x_{3}/L$, (**b**) mechanical displacement, (**c**) strain.

**Figure 7 materials-13-03115-f007:**
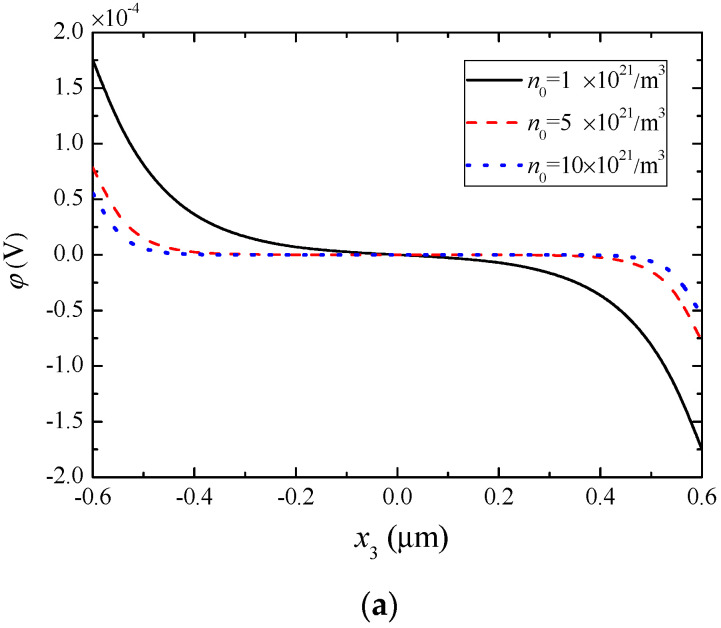
Piezoelectrically induced electric fields for different *n*_0_ when $\psi_{0}=10^{-4}\mathrm{AT}$. (**a**) Electric potential, (**b**) electric field, (**c**) electric displacement.

**Figure 8 materials-13-03115-f008:**
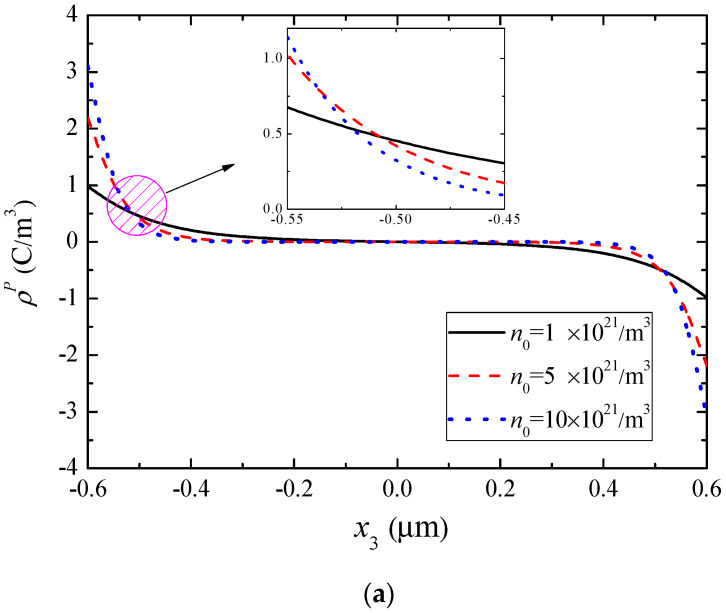
Polarization-induced charge distributions for different *n*_0_ when $\psi_{0}=10^{-4}\mathrm{AT}$. (**a**) Effective polarization charge, (**b**) electron concentration perturbation.

**Figure 9 materials-13-03115-f009:**
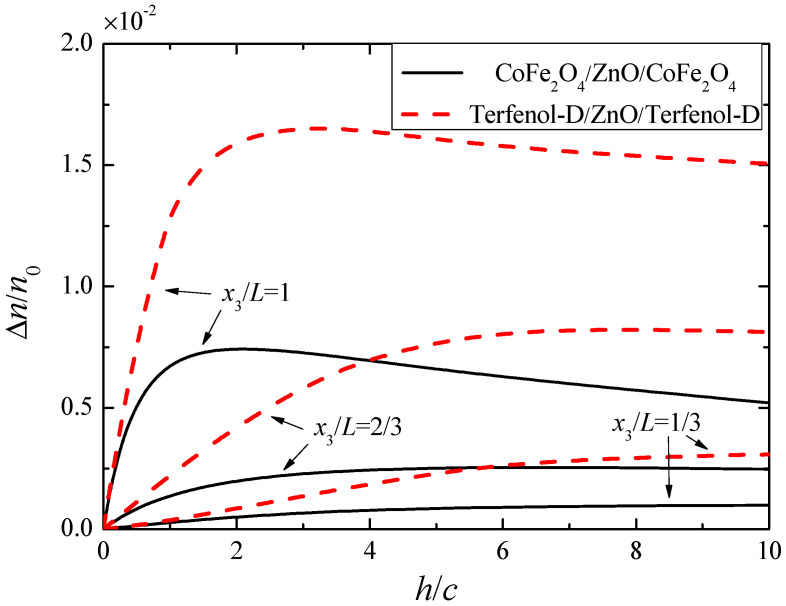
$\Delta n/n_{0}$ versus *h/c* at different locations along the rod.

**Table 1 materials-13-03115-t001:** Material properties of ZnO [50], CoFe_2_O_4_ [51], and Terfenol-D [52].

	ZnO	CoFe_2_O_4_	Terfenol-D
*c*_11_(GPa)	210	286	8.541
*c*_12_(GPa)	121	173	0.654
*c*_13_(GPa)	105	170.5	3.91
*c*_33_(GPa)	211	269.5	28.3
*e*_31_(C/m^2^)	−0.57	0	0
*e*_33_(C/m^2^)	1.32	0	0
*ε*_33_(10^−11^F/m)	8.85	9.3	5
*h*_31_(m/A)	0	580.3	−5.75
*h*_33_(m/A)	0	699.7	270.1
*μ_33_*(10^−6^Ns^2^/C^2^)	10	157	2.3

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
