# Peer review of "Magnetically Induced Carrier Distribution in a Composite Rod of Piezoelectric Semiconductors and Piezomagnetics"

_materials, 2020, doi:10.3390/ma13143115_

Round 1
Reviewer 1 Report
The manuscript presents a theoretical work on a composite rod of a piezoelectric semiconductor layer in-between two piezomagnetic layers. The rod is excited by a magnetic field, and redistribution of carriers along PS layer, strain, electric field, electric and magnetic potentials are formulated.
The paper presents an innovative and interesting idea of coupling pizomagnetic phenomenon with piezoelectric phenomenon in piezoelectric semiconductors for piezotronics.
The work is definitely worth the publication. However, I just made a minor corrections for a better presentation:
Page 2, line 55: elementary charge should be corrected, q = 1.6E-19 C.
Page 2, line 65 and 66: carrier concentrations are per volume 1/cm3 or 1/m3 which make vector D and J in these relationship have dimensions of C/cm3 and A/cm3.
Page 5, line 150, 156 and 157: similar argument might apply here. Multiplication of stress, electric displacement and magnetic induction to area will produce force, charge and magnetic flux respectively. If there is an assumption on consideration of dimensions, should be declared earlier in the manuscript.
Page 6, section 4: spacial response of various parameters along the length of the rod are calculated, but partial differential equations of part 3 includes some temporal responses which are not discussed in section 4. It would be wise to discuss time responses, since it seems to be a harmonic (dynamic) response along the rod based on these equations.
Reviewer 2 Report
The submitted manuscript concerns itself with the behavior of a multi-layered composite rod in an external axial magnetic field.
More precisely the paper presents a model proposed to explain the observed behavior: the rod exhibits a coupling between the applied magnetic field and carrier distribution, as the piezomagnetic and piezoelectric effects are combined. Calculations are performed for different materials of nowadays interest and for which there are available data and material constants.
As such, there are no actual experiments in this paper (from the abstract it's not clear that this is the case - although this is hinted, since there is no mention of any particular material), but this is not a criticism- just an observation.
Section 2 concerns itself with the presentation of the equations - in their most general form that applies here. This is a theoretical model, not necessarily complex, but it does have a lot of variables. Section 3 is about the actual 1D model that the authors study. The geometry itself is simple, a few approximations are made, and the solutions are given along with an argumentation. Section 4 presents the analytical solution to this model, while Section 4 presents the numerical results.
Overall, the article seems to add value to the specific domain, the proposed theoretical and numerical model are consistent but in the absence of any connections to experimental set-up or measurements (not necessarily made by the authors) the results and especially their conclusions could be questioned. The authors have explained (in the Numerical Results section) the discovered effects but there are not enough explanations to fully understand why and when each of the described characteristics are relevant. Also, the presentation of the results is not adequate. 8 graphs are lumped together, and are discussed in just a few paragraphs. Visually speaking, it is even hard to follow. In my opinion, 1-2 graphs at once, and the discussions around them is a good approach- this way it will be clearer what the authors are talking about.
Something else which is missing is the mention of a more concrete application (device or part of a device) that could potentially make use of the studied effect. In the introduction, only general uses are given.
However, I still think the paper shows merit, and I would like to see an improved version- as I believe the idea and results are worthy of an eventual publication.
Reviewer 3 Report
The manuscript presents a one-dimensional analytical model for a composite rod which consists of a piezoelectric semiconductor with two layers of piezomagnetic material. The numerical results show that, under the applied magnetic field, the composite rod produces extensional deformation, yielding the redistribution of mobile charge carrier. The coupled piezomagnetic and piezoelectric effects show very interesting behaviour under the magnetic field, which is the scope under investigation of the current paper. The paper is written well and could be interesting in the field of piezotronics. Therefore, I recommend the paper to be accepted after the following minor changes:
- In the Introduction, please include a literature survey of modelling efforts for such piezomagnetic-piezoelectric composite structures. Since the main novelty of this paper is in analytical modelling, the relevant studies should be referenced. Consequently, highlight the exact novelty of this work.
- Please pack the panels of Fig. 3 and Fig. 4 into one page.
- There are several typos in the text, please fix them. For instance, page 12, line 265: “Fig. (b)” should be corrected.
Reviewer 4 Report
This manuscript presents theoretical studies of a composite rod consisting of a piezoelectric layer and two piezomagnetic layers above and below the piezoelectric layer. Unfortunately, nothing more can be understood. The manuscript is completely unreadable. Although the authors did the hard work and presented interesting results, the presentation of the paper should be significantly improved.
Comments:
1) The authors should explain what are the piezoelectric and the piezomagnetic materials.
2) The authors should explain clearly what the aim of the work is: Do they apply a magnetic field that causes deformation of the piezomagnetic layer or they induce a spontaneous magnetic moment by applying physical stress? How does this affect the piezomagnetic layer? How this changes the charge distribution?
3) Where such system can be used?
4) Sentence before Eq. (1) is too long.
5) Numbering of the equations by subscripts in Eqs. (1)-(5) is not convenient.
6) Coordinate and time partial derivatives in Eqs (1)-(5) should be written as in Eq.(9).
7) The letter D is used for the electrical displacement and for carrier diffusion constant.
8) And also many other comments. The article is completely unreadable and messy.
Round 2
Reviewer 4 Report
I have already reviewed this manuscript. The authors took into account my comments and the manuscript has become much better but still needs improvement.
1) For example, the first section should begin with the words "We are considering the following system (problem) ...." and to describe the system (problem) under consideration and also provide a link to the figure where this system is depicted. Then they should write “this situation can be described by the following system of equations” and give 3-5 equations but not more. Then the Authors should write “where…” and give the definitions. Then they should give the other parts of the equations, etc.
2) The Figs with captions should be on the same page but not on two different pages.
3) There are many other comments for presentations. I recommend to authors to take a good scientific article from a good American or European journal and learn how to write such articles.
If and when the Authors will prepare their manuscript in the proper way, I will recommend
the manuscript for publication (because the scientific part of this manuscript is interesting and useful). I do not need to see the manuscript again.
Author Response
Please see the attachment。
